# Mental Healthcare through Cognitive Emotional Regulation Strategies among Prisoners

**DOI:** 10.3390/healthcare12010006

**Published:** 2023-12-19

**Authors:** Younyoung Choi, Mirim Kim, Jeongsoo Park

**Affiliations:** 1Department of Psychology, Ajou University, Suwon 16499, Republic of Korea; 2BK21 FOUR R&E Center for Psychology, Korea University, Seoul 02841, Republic of Korea

**Keywords:** mental healthcare, mental illness, cognitive emotional regulation strategy, prisoners vulnerable social group, latent profiling analysis

## Abstract

Prisoners are exposed to a deprived environment, which triggers mental illness and psychological problems. Abundant research has reported that mental illness problems, suicide, aggression, and violent behaviors occur in incarcerated people. Although the mental healthcare system for incarcerated people is emphasized, little research has been conducted due to their limited environment. In particular, the regulation of negative emotion is significantly associated with mental illness and anti-social and violent behaviors. However, mental healthcare through cognitive emotional regulation based on cognitive behavioral therapy has not been fully investigated. This study identified four different patterns in cognitive strategies for regulating negative emotions. Cognitive emotional regulation strategies (i.e., self-blame, other-blame, rumination, catastrophizing, putting into perspective, positive refocusing, positive reappraisal, acceptance, and refocus on planning) were examined and addressed their vulnerable psychological factors. We analyzed a total of 500 prisoners’ responses to the cognitive emotional regulation questionnaire (CERQ) by latent class profiling analysis. A four-class model was identified based on the responses of CERQ. In addition, the significant effect of depression on classifying the four classes was found. Furthermore, differences in the average number of incarcerations were also shown across four classes. In conclusion, Class 2 (*Negative Self-Blamer*) uses dysfunctional/negative strategies that may place the group at a high risk of psychological disorder symptoms, including depression and post-traumatic stress. Class 3 (*Distorted Positivity*) uses positive/functional strategies but seems to utilize the positive strategies in distorted manners to rationalize their convictions. Class 1 (*Strong Blamer*) and Class 4 (*Moderator Blamer*) showed similar patterns focused on the “other-blame” strategy for regulating negative emotion, but they are at different levels, indicating that they attribute incarceration to external factors. These findings provide useful information for designing mental healthcare interventions for incarcerated people and psychological therapy programs for clinical and correctional psychologists in forensic settings.

## 1. Introduction

Sykes’s research [1] suggested that prisoners experience five types of deprivation in prison, including freedom, goods and services, sexual relations, autonomy, and security. Prisoners are socially isolated, receive poor-quality materials and services, are prohibited from contact with the opposite sex, and must passively follow the rules. The incarcerated environment can cause mental illnesses such as depression and anxiety and anti-social behaviors such as suicide, aggression, and violence. In fact, incarcerated people suffer from mental illness with maladaptive psychological characteristics such as anxiety and depression, as well as irrational beliefs and erroneous biases [2]. The Prison Policy Initiative (2022) reported that over 40% of people suffer from mental health disorders and experience subsequent depression and bipolar disorder due to the incarcerated environment [3,4]. In addition, the Prison Policy Initiative (2022) estimated that a prisoner is three times more likely to die from suicide than the general population [5]. 

The features of incarceration, such as disconnection from family, loss of autonomy, boredom and lack of purpose, and unpredictability of surroundings, are linked to negative mental health outcomes [6]. In addition, researchers at the University of Georgia found that people incarcerated from home were more likely to experience depression in 214 state prisons [7]. Furthermore, the number of incarcerations of offenders who were released from prison is significantly related to poor mental health in prison. Specifically, the number of incarcerations of offenders who suffer from poor mental health in prison is higher than in the general population [8,9]. However, research has also reported that prisoners have barriers to mental health treatment [10]. Mental healthcare and related professionals are rarely available in prisons. Furthermore, prisoners find it difficult to access mental healthcare in general [10]. 

The experience of continuous exposure to such an incarcerated environment evokes a variety of negative emotions. Mostly, negative emotions are the basis for triggering various mental illnesses and behavioral problems. Consequently, incarceration can worsen depression and last a long time after leaving prison [6]. In addition, the prison environment has various elements that cause distorted cognitive thinking. For example, Seo et al. [11] reported that murderers in prison possess distorted cognitive thinking processes, which lead to aggressive behaviors and suicidal thoughts. Although the mental healthcare system for incarcerated people is emphasized, few research studies have been conducted due to their limited environment. Previous research suggested that mental healthcare in terms of the ability to regulate negative emotions acts as a buffer for triggering various mental illnesses and behavioral problems such as suicidal behaviors, violence, depression, etc. [12,13,14,15]. Although prisoners’ psychological problems due to the prison environment have been consistently reported as a major issue, negative emotions and distorted cognitive thinking processes among prisoners, have not been fully investigated. In addition, mental healthcare intervention regarding cognitive–emotional regulation has not been fully addressed. 

Emotional regulation refers to the ability to manage one’s emotional experiences and behaviors, particularly in situations of intense emotional arousal [16]. Emotion dysregulation is related to mental health, wherein negative emotions highly influence poor mental health. In addition, research found that trauma-exposed community individuals have shown a relationship between emotion dysregulation and mental health [17]. Furthermore, difficulties in regulating emotion have strong associations with the characteristics of psychopathy, including lack of empathy, impulsivity behaviors, and criminal tendencies [12,13,14,15,16]. 

Research has also reported that the characteristics of psychopathy are related to criminality and incarceration [16,17,18,19,20]. In a similar vein, Garofalo, Neumann, and Velotti [20] found that emotion dysregulation plays a significant mediation role in the association between psychopathy and aggressive behaviors. Preston and Anestis [21] also found that emotion dysregulation mediated the association between self-centered impulsivity traits of psychopathy and reactive aggression. In addition, Garofalo et al. [22] reported impairments in emotional regulation influence anger in both community and offender samples. Consequently, mental healthcare in forensic settings focus on correlational interventions and psychological therapy programs related to emotional regulation for reducing anti-social and aggressive behaviors [21]. 

Garnesfski, Kraaij, and Spinhoven [22] suggested that people choose internal or cognitive–emotional regulation methods as they grow. Therefore, cognitive strategies for regulating emotion may explain affective and cognitive mechanisms that underlie the externalizing and anti-social behaviors of adults. Garnefski, Kraaij, and Spinhoven [22] proposed a scale for CER (CERQ: Cognitive Emotion Regulation Questionnaire) in order to assess different types of cognitive strategies for regulating emotion under stressful events. Cognitive–emotional regulation (CER) refers to a cognitive process of managing emotions to deal with stressful events [22]. The CERQ measures nine types of CER strategies, including self-blame, other-blame, rumination, catastrophizing, putting into perspective, positive refocusing, positive reappraisal, acceptance, and refocusing on planning. These strategies are used for managing negative emotions in response to stressful or threatening events. In more detail, self-blame involves putting the blame on oneself for what has been experienced, while other-blame involves attributing the blame to the environment or others. Rumination is defined as repeatedly focusing on thoughts about the feelings associated with a negative event. Catastrophizing is characterized by thoughts that explicitly emphasize the severity of what has been experienced. Putting it into perspective involves downplaying the seriousness of the event and emphasizing its relative significance when comparing it to other events. Positive refocusing involves thinking about joyful and pleasant issues instead of the actual negative event. Positive reappraisal involves creating a positive meaning for the event in terms of personal growth. Lastly, acceptance is the act of accepting what has been experienced. Refocus on planning involves thoughts of resigning yourself to what has happened and reorganizing and making a plan about the events. 

Abundant research regarding emotional regulation has reported robust associations with mental health. In addition, a feature of depressive disorder commonly includes cognitive changes that significantly influence the individual’s function [23]. However, the identification of different types of cognitive strategies for regulating emotion under stressful situations among incarcerated people has not been fully investigated. 

In addition, different patterns of cognitive strategies in emotional regulation may be related to the frequent anti-social behaviors and crimes. Therefore, the investigation regarding the association between different types of CER strategies can provide useful information for designing a mental health intervention and psychological therapy program for caring for mental illness among incarcerated people in forensic settings. 

### The Present Study

The main aim of this study is to identify different patterns of CER strategies among prisoners and to see how depressive disorder would impact the classifications of CER strategies. In addition, the study examines how much the different patterns can explain the number of incarcerations as the manifest variable of frequent anti-social behaviors. To achieve this, we first analyze the prisoners’ responses to CERQ, which consists of nine different emotional strategies in response to negative events. Latent class profiling analysis is conducted to identify latent patterns of CER strategies. Afterward, Wald tests are used to examine the significant differences in the average number of incarcerations among the identified classes. 

## 2. Method

### 2.1. Design and Setting

A total of 521 prisoners were selected from a prison in Pusan, South Korea, which has a moderate security level (S4). The psychological interviews were conducted by four doctoral-level clinical psychologists; basic psychiatric symptoms, including depression and anxiety, were evaluated for all participants. Additionally, a simple assessment of cognitive functioning, including memory, attention, concentration, and executive functioning, was conducted prior to the survey. Participants with severe cognitive impairment were excluded from the study. All participants were informed of the purpose and anonymity of the research. Each of the five prisoners conducted the survey in a privately blocked room. A USD 25 payment was provided to each participant. Finally, the responses of 500 prisoners were analyzed. This study received approval from the Human Subjects Review Committee at Donga University (2-1040709-AB-N-01-202001-BR-003-04). 

### 2.2. Participant

After removing missing values and careless responses, 500 samples were analyzed in this study. The participants’ basic demographic information and psychological characteristics were computed (Table 1). Their ages ranged from 16 to 75 years (Mean = 46.69, SD = 11.59). The prisoners were incarcerated for a variety of convictions, including homicide (29.8%), violent offenses (15.4%), sexual violence offenses (30.0%), drug-related crimes (4.0%), property offenses (17.2%), and others (3.6%). Regarding education levels based on their last offense, 17.2% had only completed primary school (Year 6), 25.8% completed middle school, 44.0% finished senior high school, and 13.0% completed tertiary or above education. In terms of employment, 57% of the participants had a full-time job, 29% had a part-time job, and 14% were unemployed before being incarcerated. We also examined the basic psychological characteristics of the subjects, which were measured using the PHQ9 scale (Mean = 30.64, SD = 2.67).

### 2.3. Measures

#### 2.3.1. Cognitive Emotional Regulation Questionnaires 

CER was assessed using the Korean version of the Cognitive Emotion Regulation Questionnaire (CERQ), which was validated by Ahn, Lee, and Joo [24]. The concept and CERQ were originally proposed by Garnefski and Kraaij [25]. The CERQ consists of 64 items measuring nine sub-factors of self-blame (SB), acceptance (ACP), rumination (RM), positive refocusing (PRF), positive reappraisal (PRA), putting into perspective (PIP), catastrophizing (CAT), blaming others (BO), and refocus on planning (ROP). Participants responded to each item on a five-point Likert scale ranging from 1 (Almost Never) to 5 (Almost Always). Cronbach’s α of the total CERQ was 0.87. Cronbach’s α of each sub-factor is presented on the diagonal of Table 2, along with correlation values on the off-diagonal.

#### 2.3.2. Number of Incarcerations

The average number of incarcerations was 2.39 (Min = 1, Max = 15), as shown in Table 2.

### 2.4. Analysis Procedures

The analyses were conducted based on three main hypotheses in the current study. The first hypothesis posited that the responses of prisoners would be classified into different latent classes based on their cognitive–emotional regulation (CER) strategies. The second hypothesis is that depression, which is an important factor in mental health, would have an impact on the classification of the latent classes given CER strategies. Finally, the number of incarcerations would vary across the different latent classes of CER strategies. To test these hypotheses, we applied latent class profiling (LCP) analysis to the prisoners’ responses to the CERQ and tested multinomial logistic regression to see the effect of depression on the classification of latent profiles. In addition, the Wald test was conducted to compare the four latent classes in terms of the average number of incarcerations. All analyses were conducted using *Mplus* 8.7 [26]. 

The LCP assumed unobservable latent sub-populations (i.e., latent classes) underlying the observed data [27]. In this study, we assumed that the prisoners had unique response patterns on CERQ, depending on the latent class to which they belonged. The LCP was based on the three-step method to lessen estimation bias resulting from the inclusion of the distal outcome (i.e., # of incarcerations) and to take the measurement error into account [28,29]. Therefore, two research models were analyzed in separate steps: (a) participants were classified into latent CER strategy classes based on their CERQ responses, and (b) differences in average number of incarcerations were examined across the latent classes.

## 3. Results

### 3.1. Identifying Latent Classes

Following Nylund et al.’s [30] recommendation, the bootstrap likelihood ratio test (BLRT) [31] and BIC [32] were used to decide the number of latent classes. BLRT assessed the improvement in model fit between neighboring class models by adding an additional class (i.e., comparison between *k* − 1 and *k* class models). If BLRT was significant, *k* classes would be preferred over the *k* − 1 classes. In BIC, a smaller value would indicate a better-fitting model for classification. We also considered each class size (i.e., the number of individuals in the class) and entropy as indicators of the model’s fit. If a particular class had only a few assigned individuals, the model was not preferred due to interpretability issues. We evaluated the quality of classification based on entropy, counts of individuals in each class, and the meanings of the classification to avoid data-driven decision-making. The entropy, ranging from 0 to 1, would indicate a better classification as it increased, and enough class sizes are necessary for the interpretation [33]. Table 3 presents fit indices across diverse class models. Based on the above criteria, a four-class model was selected and used for further analysis. Although a five-class model had the smallest BIC and a significant BLRT, one of the classes had a small number of individuals (*n* = 18) contemplating the meaningful interpretation of the corresponding class [27,33]; therefore, we decided on the four-class model, considering its interpretable latent classes and adequate entropy value. 

Figure 1 displays the CERQ patterns of the four-class model, indicating the sub-factor scores, while Table 4 provides the corresponding demographic information for each class. Looking at Figure 1, we found that Class 1 and Class 4 have a particular characteristic of blaming others. These two groups had similar patterns but at different levels, in which Class 1 showed the use of a strategy exhibiting strong blame towards others, while Class 4 exhibited moderate blame towards others. Class 4 had a higher score of BO than Class 1, but Class 1 had the noticeably highest score of BO among other scores. On the other hand, Class 4 reported higher scores in other functional strategies than Class 1. 

Class 2 consisted of a group of negative self-blamers. They were more likely to have CER strategies involving negative perspectives and less likely to blame others compared to other classes. Moreover, Class 2 reported higher scores in terms of dysfunctional strategies than the other classes. 

Lastly, Class 3 represented a group exhibiting distorted positivity group. Participants in this class showed a similar pattern to Class 2 in terms of SB, ACP, and RM, but they had higher scores in PRF, PRA, PIP, and a lower score in CAT. Furthermore, both Class 3 and Class 4 reported higher scores in PRF, PRA, and PIP than in Class 2. 

Taken together, individuals in Class 1 (strong blamers toward others) generally showed lower scores across all nine sub-factors compared to the other classes, with the highest BO sub-factor scores. Class 4 (moderate blamers to others) was similar to Class 1, but its scores on all sub-factors were higher than those of Class 1. In other words, people in C4 generally utilized all strategies. The other two classes, Class 2 and Class 3 had unique patterns. Although Class 2 had higher scores in all nine strategies than Class 1, the SB, ACP, RM, and CAT scores were higher than PRF, PRA, PIP, and BO. Notably, individuals in Class 2 had the highest CAT score among all the classes, indicating a greater tendency to employ negative strategies of SB, RM, and CAT. In addition, Class 2 showed a unique pattern with larger bumps compared to the other classes. Individuals within Class 2 had the highest scores of SB, RM, and CAT, reflecting negative perspectives towards the unpleasant situation in prison; however, they have sharp drops at PRF, PRA, and PIP related to the positive perspectives and BO. Class 3 generally used all strategies, with relatively higher use of ACP, PRF, PRA, PIP, and RFP and less use of CAT and BO, which represented negative/dysfunctional strategies. Although Class 3 employed relatively more adaptive emotional strategies than the other classes, considering the situation where prisoners were convicted and incarcerated, these strategies may not be truly adaptive emotional strategies but rather distorted positivity.

### 3.2. Multinomial Logistic Regression Analysis 

Based on the literature [6,23], we considered depression as an important predictor to differentiate CER strategies and examined the effect of depression on the classification of the four classes using a multinomial logistic regression analysis. The results are presented in Table 5. As Class 2 was set as a reference class for the analysis, the regression effect indicates the extent to which the depressive disorder categorized the individuals into a specific latent class rather than into Class 2. The depression was a significant predictor to differentiate prisoners of Class 1/Class 3 from Class 2. The odds ratios for C1 and C3 were 0.80 and 0.85, respectively, indicating the decreased odds of classification into Class 1 and Class 3 by 20% and 15% compared to Class 2, when the level of depression increased.

### 3.3. Number of Incarcerations Comparisons across Classes

Given the four-class model, we investigated how the number of incarcerations varied across these classes, controlling the effect of depression. Table 6 presents the average number of incarcerations for each class: C1 (2.22); C2 (1.71); C3 (2.49); and C4 (2.46). Prisoners in Class 2, which was characterized by a higher level of CAT and a lower level of BO than the other classes, reported the lowest number of incarcerations. In addition, the proportion of first-time prisoners in Class 2 (71.1%) was the highest among the four classes: C1 (44.4%); C3 (45.2%); and C4 (50.2%). We conducted Wald tests to test statistical differences in the average number of incarcerations between classes and provided the test result in Table 6. As shown in the table, Class 2 showed statistically significant differences in the number of incarcerations compared to Class 3 and Class 4. The negative estimates for the Wald test indicate that the latter class had a higher number of incarcerations than the former; therefore, Class 3 and Class 4 had a higher average number of incarcerations than Class 2. 

The identified patterns of CER strategies in the current study explain the relatively lower number of incarcerations of Class 2 compared to Class 3 and Class 4. Class 3 had higher scores in PRF, PRA, and PIP but a lower score in CAT than Class 2. On the other hand, Class 4 had higher scores in PRF, PRA, PIP, and BO compared to Class 2, while they reported lower scores in SB, ACP, RM, and CAT. In conclusion, both classes commonly showed higher scores in PRF, PRA, and PIP, and a lower score in CAT. Therefore, the prisoners with more records of incarcerations reported higher scores in PRF, PRA, and PIP, and lower scores in CAT. However, the other comparisons were not statistically significant.

## 4. Discussion

The main aim of this study was to identify latent classes representing different patterns of CER strategies among prisoners and examine how these patterns explained incarcerations as an indicator of anti-social behaviors. Initially, we analyzed the prisoners’ responses to CERQ, which consists of nine different strategies for regulating emotion in negative and stressful situations. LCP analysis was conducted to identify latent classes regarding different patterns of CER strategies, and the effect of depression on categorizing prisoners into specific latent classes was examined. Subsequently, Wald tests were used to examine whether the number of incarcerations varied across the latent classes and to see how the different CER strategies explained the incarcerations. 

The present study identified four latent classes based on the scores of CER strategies. First, Class 2, referred to as “Negative Self-blamer,” consisted of 45 prisoners, accounting for 9% of the total sample. This class was more likely to adopt negative/dysfunctional strategies (i.e., SB, RM, and CAT) for regulating emotion under stressful events. These individuals may blame themselves for incarceration, experience negative thoughts and emotions, and perceive their incarceration experiences negatively. Given the associations between negative/dysfunctional strategies and emotional dysregulation, depression, or post-traumatic stress [8,12], Class 2 may be at a high risk of psychological disorder symptoms. The current study found that Class 2 experienced a significantly lower number of incarcerations compared to Class 3 and Class 4, which reported higher scores in PRF, PRA, and PIP strategies. Previous research has indicated that anti-social and violent behaviors may occur as a means to cope with negative emotions [26]. Therefore, cautious interventions addressing psychological symptoms may help prevent suicide in prison and the likelihood of future incarceration within this group. 

Second, Class 1 (*n* = 36) and Class 4 (*n* = 231) showed behavior referred to as “Moderate blamer toward others” and “Strong blamer toward others”, respectively, and showed similar patterns of CER strategies focused on BO, but at different levels. These two classes accounted for 53.4% of the total sample, indicating that a significant proportion of prisoners attribute their anti-social behaviors or incarceration to external factors. This finding aligns with previous research showing a lack of self-criticism among a considerable number of prisoners [23]. It would be helpful to challenge their attribution and elicit self-directed behavioral changes in correction programs. Considering that placing high-risk and low-risk prisoners in the same group reduced program effects [27,28], it would be more effective to implement separate interventions in terms of emotional regulation patterns. In addition, the individuals in Class 1, with higher scores in BO and CAT and lower scores in the other sub-factors, were most likely to have low motivation to participate in correctional programs voluntarily. Therefore, behavioral approaches, such as contract agreements that enforce attendance through reduced control or incentives, may be helpful in increasing therapeutic effectiveness. 

Third, Class 3 (*n* = 188) referred to as “Distorted Positivity,” representing 37.6% of the total sample. This class used more positive/functional strategies, such as ACP, PRF, PRA, PIP, and RFP, compared to negative/dysfunctional strategies, such as CAT. Class 4 also reported a similar pattern to Class 3; both classes reported higher scores in PRF, PRA, and PIP compared to Class 1, as well as lower scores in CAT. However, Class 3 and Class 4 differed in terms of blame strategies; SB was more prevalent in Class 3, while BO was more prevalent in Class 4. Positive/functional strategies are generally helpful for psychological symptoms in the normal population. However, prisoners in Class 3 and Class 4 seemed to utilize these positive strategies in distorted manners to rationalize their convictions and crimes, given the statistically higher average number of incarcerations compared to Class 2. Although Class 3 and Class 4 experienced repeated incarcerations, they appeared to be well-adjusted in prison through their adaptive/functional emotional regulation skills. 

## 5. Implications and Limitations

The findings of this study have several implications. First, to our knowledge, this is the first study to investigate different patterns of Cognitive–Emotional Regulation (CER) strategies among prisoners, providing insights into their intricate emotional and cognitive mechanisms for managing negative emotions. Moreover, recognizing the association between incarceration and poor mental health of offenders in prison [7], investigating patterns of emotional dysregulation as precursors to psychopathy [15,16,20,34,35], aggression [20,21], or depression [36,37] may serve as grounded evidence to develop rehabilitation intervention targeting mental health. 

Second, given that the amalgamation of high-risk prisoners and low-risk prisoners in the same intervention group resulted in diminished effectiveness [38], our findings emphasize the importance of tailed interventions based on risk levels. Despite Class 2 (i.e., Negative Self-blamer) showing a lower incarceration rate compared to Class 3 and Class 4, this group is more susceptible to experiencing psychological disorder, potentially leading to suicide. One recent study suggests that self-blame is a more significant factor in predicting suicidal ideations than factors such as age, gender (male), and high perceived stress [39]. Moreover, while positive/functional strategies are typically recommended in normal populations for their psychological symptoms [40], our results indicate that Class 3, with higher use of adaptive emotional strategies, indicates repeatedly incarcerated for offenses. In addition, Class 1 indicates lower motivation to participate in correctional programs. Thus, recognizing each class suggests the necessity for different approaches intervening in each group.

Third, significantly different levels of incarceration across latent classes driven by CER strategies among prisoners highlight the importance of individual differences in CER when designing correction programs. Despite the general efficacy of cognitive–behavioral approaches in reducing criminal behaviors among offenders [41], research indicates differing responsiveness to interventions among offenders [38,42,43]. In terms of correctional intervention effectiveness, understanding how individuals regulate their emotions leading to incarceration can inform program design adjusting to specific cognitive and emotional processes. For example, improvement in terms of emotional and cognitive regulation at a different level was associated with juvenile recidivism [44]. Consequently, rehabilitation programs within prisons could benefit from personalized interventions imparting adaptive emotional regulation skills, with the aim of mitigating the risk of post-release anti-social behaviors. Thus, tailoring interventions to specific patterns regarding cognitive emotion strategies may enhance their effectiveness and reduce the risk of anti-social behaviors and recidivism. 

Despite its contribution, this study has several limitations. First, the use of self-report measures may bring related bias and inaccuracy. Future research could benefit from incorporating objective measures or observational data to provide a comprehensive understanding of these strategies. Second, the sample in this study consisted of a specific group of prisoners, which limits the generalizability of the findings to other prisoners or contexts. Future studies with diverse prisoners or different contexts would strengthen and expand upon these results. Lastly, the cross-sectional study design prevents the establishment of causal relationships between CER and frequent anti-social behaviors. Further studies are needed to examine the temporal relations and predictive validity of these patterns in relation to reoffending behaviors. 

## Figures and Tables

**Figure 1 healthcare-12-00006-f001:**
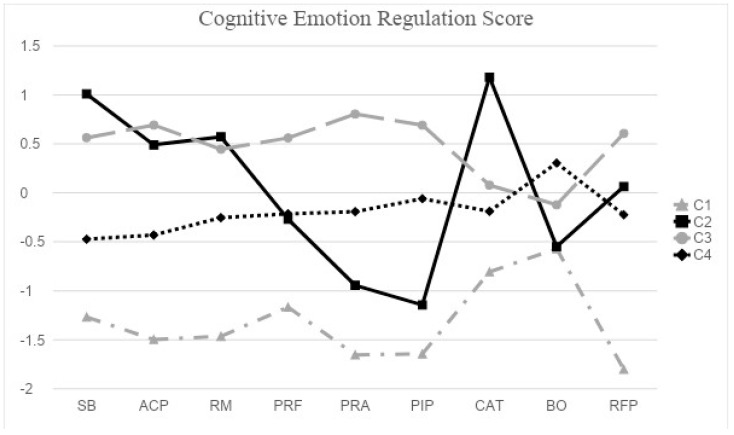
Cognitive emotion regulation scores across four classes. Note: SB—Self-blame; ACP—Acceptance; RM—Rumination; PRF—Positive refocusing; PRA—Positive reappraisal; PIP—Putting into perspectives; CAT—Catastrophizing; BO—Blaming others; RFP—Refocus on planning.

**Table 1 healthcare-12-00006-t001:** Demographic information and psychological characteristics of the participants.

Demographic Information	Categories	*N*	%
Criminals	Homicide	149	29.8
	Violent Offenses	77	15.4
	Sexual Violence Offenses	150	30.0
	Property Offenses	86	17.2
	Drug-Related Crimes	20	4.0
	Others	18	3.6
Years of Education	Below Primary School	12	2.4
	Primary School	73	14.6
	Middle School	133	26.6
	High School	221	44.2
	Above College	60	12
Job Status	Full-time Jobs	285	57.0
	Part-time Jobs	145	29.0
	Unemployed	70	14.0
Total		500	
		**Mean**	**SD**
Age		46.69	11.59
# of incarcerations		2.39	2.20
Depression	PHQ9	30.64	2.68

**Table 2 healthcare-12-00006-t002:** Correlations, mean, and standard deviation of cognitive emotion regulation strategies.

CER Strategy	1	2	3	4	5	6	7	8	9
Self-blame	0.80								
Acceptance	0.55 **	0.64							
Rumination	0.47 **	0.41 **	0.66						
Positive refocusing	0.28 **	0.33 **	0.19 **	0.75					
Positive reappraisal	0.26 **	0.47 **	0.29 **	0.48 **	0.74				
Putting into perspectives	0.23 **	0.40 **	0.23 **	0.38 **	0.64 **	0.59			
Catastrophizing	0.41 **	0.19 **	0.58 **	−0.02	−0.05	−0.08	0.78		
Blaming others	−0.27 **	−0.10 *	0.11 *	−0.06	0.09 *	0.13 **	0.17 **	0.74	
Refocus on planning	0.32 **	0.45 **	0.36 *	0.40 **	0.58 **	0.43 **	0.03	0.05	0.77
**Mean**	13.80	11.07	13.10	13.74	13.99	13.75	10.76	9.77	15.87
**SD**	3.78	2.55	3.20	3.47	3.55	3.04	3.99	3.40	3.05

Note. CER—cognitive emotional regulation; diagonal elements indicate Cronbach’s α; ** *p* < 0.01; * *p* < 0.05.

**Table 3 healthcare-12-00006-t003:** Fit indices for identifying latent classes.

Fit Index	2 Classes	3 Classes	4 Classes	5 Classes
BIC	12319.27	12100.90	11937.32	11865.87
BLRT	625.19 **	280.52 **	225.72 **	133.59 **
Class sizes	143/357	22/242/236	36/45/188/231	18/105/224/47/106
Entropy	0.78	0.81	0.84	0.81

Note. ** *p* < 0.01.

**Table 4 healthcare-12-00006-t004:** Demographic information and psychological characteristics across classes.

Demographic Information	Categories	Class 1	Class 2	Class 3	Class 4
		Strong Blamer	Negative Self-Blamer	Distorted Positivity	Moderate Blamer
		Frequency (%)			
Criminals	Homicide	11 (30.6)	23 (51.1)	51 (27.1)	64 (27.7)
	Violent Offenses	4 (11.1)	6 (13.3)	33 (17.6)	34 (14.7)
	Sexual Violence Offenses	16 (44.4)	12 (26.7)	42 (22.3)	80 (34.6)
	Property Offenses	3 (8.3)	2 (4.4)	38 (20.2)	43 (18.6)
	Drug-Related Crimes	-	1 (2.2)	14 (7.4)	5 (2.2)
	Others	2 (5.6)	1 (2.2)	10 (5.3)	5 (2.2)
Educational Years	Below Primary School	-	1 (2.2)	7 (3.7)	4 (1.7)
	Primary School	8 (22.2)	9 (20.0)	23 (12.2)	33 (14.3)
	Middle School	9 (25.0)	11 (24.4)	37 (19.7)	76 (32.9)
	High School	17 (47.2)	19 (42.2)	93 (49.5)	92 (39.8)
	Above College	2 (5.6)	5 (11.1)	27 (14.4)	26 (11.3)
Job Status	Full-time Jobs	21 (58.3)	28 (62.2)	111 (59.0)	125 (54.1)
	Part-time Jobs	9 (25.0)	9 (20.0)	53 (28.2)	67 (29.0)
	No Jobs	6 (16.7)	8 (17.8)	24 (12.8)	39 (16.9)
Total		36	45	188	231
		**Mean (SD)**			
Age		43.32 (12.46)	51.16 (11.30)	47.74 (10.46)	45.44 (12.12)
# of recidivism		2.22 (1.82)	1.71 (1.58)	2.49 (2.28)	2.46 (2.27)
Depression	PHQ9	29.94 (2.62)	31.49 (2.46)	30.35 (2.63)	30.83 (2.72)
Anxiety	STAI-S	37.75 (6.59)	40.56 (6.53)	42.97 (6.04)	40.37 (6.87)
	STAI-T	38.17 (7.63)	40.11 (6.03)	43.95 (5.70)	41.12 (6.38)

**Table 5 healthcare-12-00006-t005:** Multinomial logistic regression with depression.

Class Comparison	Estimate	S.E.	*p*-Value	Odds Ratio
Depression → C1	**−0.22**	0.09	0.01	**0.80**
Depression → C3	**−0.16**	0.06	0.01	**0.85**
Depression → C4	−0.08	0.06	0.10	0.91

Note. Reference class = C2; bold indicates significant test statistics.

**Table 6 healthcare-12-00006-t006:** Wald tests across latent classes for the number of incarcerations.

Class Comparison	Estimate	S.E.	*p*-Value
C1 vs. C2	0.50	0.38	0.18
C1 vs. C3	−0.27	0.34	0.43
C1 vs. C4	−0.24	0.34	0.48
C2 vs. C3	**−0.78**	0.29	0.01
C2 vs. C4	**−0.74**	0.28	0.01
C3 vs. C4	0.03	0.23	0.88

Note. Bold indicates significant test statistics.

## Data Availability

The derived data supporting the findings of this study are available from the corresponding author on request.

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
