# Peer review of "Mental Healthcare through Cognitive Emotional Regulation Strategies among Prisoners"

_healthcare, 2023, doi:10.3390/healthcare12010006_

Round 1

Reviewer 1 Report

Comments and Suggestions for Authors This paper examines (cognitive) strategies to regulate negative emotions in a sample of (Korean) prisoners. Using Latent Class Profiling in conjunction with the Cognitive Emotion Regulation Questionnaire, it suggests a four-fold model of regulation.   It situates the study within some literature, and offers a conventional description of the methods. The results are also clearly grounded in evidence and the paper appears to make a contribution to knowledge that is of potential interest to the readership of the journal.    I would make two comments for further consideration before publication is approved.   The first concerns the use of literature. The literature review is quite general, the methodological literature cited is a little sparse (see below), and although some implications are discussed, they are not grounded in any literature. This has the effect of weakening the connection between the paper and the wider field. I suspect you could tighten the literature review a little, make some reference to the technical literature on LCP in the methods, and add some relevant references to the implications section. This would help to enhance the visibility of the paper.    Second, a better case could be made as to why entropy was the preferred measure for the four class model, when the BIC and BLRT measures indicate a five class model. The overall sample size appears large enough to support the use of these tests, so why entropy? Citing some supporting (technical) literature here would strengthen the paper, particularly as the entire content of the paper is based on this decision.      Comments on the Quality of English Language

The paper could benefit from thorough proofreading as there are some minor stylistic anomalies. 

Author Response

We appreciate your suggestions and comments. We revised our manuscript and added more detailed literature. Please take a look at the attached file for the details.

Reviewer 2 Report

Comments and Suggestions for Authors

It is a methodologically correct work and its results are relevant in relation to the subject that has been investigated. However, we note a series of problematic aspects in both the theoretical review and the methodological design used. 

With regard to the background information reviewed, a lack of bibliographical citations specifically referring to the mental health problems of the prison population is detected (only 2 and 3 citations are provided, not too many of which are related to the subject). It seems logical that if the title of the paper specifies the topic of mental health, it should be based on a minimum number of articles that review this issue. We recommend that some previous studies on this subject be provided.

Furthermore, when studies on emotional regulation with prisoners are collected, they are limited exclusively to studies with subjects who present psychopathic characteristics, which seems logical in the research population, but there is a lack of work that analyses mental health problems related to anxiety, depression and suicides, and their relationship with emotional regulation.

In fact, we consider it necessary to contribute studies that analyse the variable of psychopathy and its relationship with the mental health of prisoners, in the interest of analysing the influence of this personality disorder on the emotional adjustment of people who have been deprived of their liberty for having committed criminal acts.

In terms of methodology, we consider that the inappropriateness of the title of the paper is emphasised by establishing the mental health of prisoners as its central core, since the influence of isolated latent patterns on the variables of depression and anxiety assessed by means of the instruments applied (PHQ9, STAI-S and STAI-T) is not analysed at all. We believe that in order to give coherence to the subject matter of the study, some descriptive and predictive analysis should be carried out to analyse the possible relationship between the latent classes and the emotional maladjustments of the sample studied.

Finally, the inappropriateness of the statement "CER strategies explained recidivism" should be pointed out, since the statistical analysis used (Wald test) only describes differences. In order to reach this conclusion, some kind of predictive analysis should be carried out to isolate the possible influence.

Author Response

(The authors gave the same response as above.)

Round 2

Reviewer 2 Report

Comments and Suggestions for Authors

The improvements made to the paper are in line with the indications given by the reviewer. The only thing to note is that it is not considered necessary to modify the title of the paper, introducing the concept of depression, as this would imply focusing the subject matter on this variable, leaving other psychopathic variables that are also studied in the research in second place. Depression has already been adequately covered in the content of the paper, thus broadening the perspective of analysis. We therefore recommend keeping the title as in the first version, so that it has a more general orientation, focusing on the topic of interest of mental health. 

Author Response

We appreciate you giving us specific comments and reading our revised manuscript thoroughly.

Following your suggestion, the revised manuscript keeps the original title "Mental Healthcare through Cognitive Emotional Regulation Strategies among Prisoners".

Thank you for your feedback again.